# Reduction in maternal mortality ratio varies by district in Sidama National Regional State, southern Ethiopia: Estimates by cross-sectional studies using the sisterhood method and a household survey of pregnancy and birth outcomes

Aschenaki Zerihun Kea[1,2]*, Bernt Lindtjorn[1,2], Achamyelesh Gebretsadik Tekle[1], Sven Gudmund Hinderaker[2]

1 School of Public Health, College of Medicine and Health Sciences, Hawassa University, Hawassa, Ethiopia, 2 Centre for International Health, University of Bergen, Bergen, Norway

* aschenakizer@yahoo.com

**Data Availability Statement:** All relevant data are within the manuscript.

## Abstract

### Background

Few studies assess the magnitude, variations, and reduction of maternal mortality at a lower administrative level. This study was conducted to estimate the life time risk (LTR) of maternal death and the maternal mortality ratio (MMR) and assess the reduction in MMR.

### Methods

This is a population-based cross-sectional study conducted in six districts of Sidama National Regional State, southern Ethiopia, from July 2019 to May 2020. The study was conducted with men and women aged 15–49 years. By creating a retrospective cohort of women of reproductive age, we calculated the LTR of maternal mortality and approximated the MMR using the total fertility of the rural Ethiopian population. Variations in maternal mortality was assessed based on characteristics of the respondents, like age, sex, and the districts where they lived. Reduction in MMR was examined using the estimates of the sisterhood method and the 5-year recall of pregnancy and birth outcome household survey.

### Results

We analysed 17374 (99.6%) respondents: 8884 (51.1%) men and 8490 (48.9%) women. The 17,374 respondents reported 64,387 maternal sisters. 2,402 (3.7%) sisters had died; 776 (32.3%) were pregnancy-related deaths. The LTR of maternal death was 3.2%, and the MMR was 623 (95% CI: 573–658) per 100,000 live births (LB). The remote district (Aroresa) had a MMR of 1210 (95% CI: 1027–1318) per 100,000 LB. The estimates from male and female respondents were not different. A significant reduction in MMR was observed in

**Funding:** This study was funded by Grand Challenges IDRC Canada through the Liverpool School of Tropical Medicine and REACH Ethiopia, a non-profit organization. The funders had no role in study design, data collection and analysis, decision to publish, or preparation of the manuscript.

**Competing interests:** The authors declare that they have no competing interests.

districts located near the regional centre. However, no reduction was observed in districts located far from the regional centre.

## Conclusions

The high MMR with district-level variations and the lack of mortality reduction in districts located far from the centre highlight the need for instituting interventions tailored to the local context to save mothers and accelerate reductions in MMR.

## Introduction

The government of Ethiopia has been implementing interventions aiming to reduce maternal mortality since the commencement of the Millennium Development Goal (MDG) [1], and the maternal deaths in the country fell by 61% during the MDG period. Despite these achievements, in 2017, there were an estimated 14000 maternal deaths, with a corresponding maternal mortality ratio (MMR) of 401 per 100,000 live births (LB) [2].

Currently, maternal deaths are not routinely tracked using vital registration in Ethiopia [3, 4]. In certain areas, maternal deaths are tracked through studies conducted in special settings like demographic surveillance sites [5] and hospitals [6]. The data used for monitoring maternal mortality in the country largely comes from Demographic and Health Surveys (DHS) [7] and United Nations agency [2] reports conducted periodically at the national level. However, national level studies do not provide data on the magnitude and variations of maternal deaths at the sub-national and district levels, useful for program planning and monitoring maternal mortality at the lower administrative level.

A recent population-based household survey conducted in Sidama Regional State identified a high MMR with district-level variations. The overall MMR of the region, reported by this study, was 419 (95% CI: 260–577) per 100,000 LB, but Aroresa district (the remote district) had a MMR of 1142 (95% CI: 693–1591) deaths per 100,000 LB [8].

We carried out a population-based household survey employing the indirect sisterhood method to measure maternal mortality in Sidama National Regional State, southern Ethiopia. The indirect sisterhood method was part of a larger maternal mortality survey aimed at measuring maternal mortality in Sidama National Regional State using a retrospective 5-year recall period of pregnancy [8]. The larger study also included measuring physical access to skilled delivery using the Geographic Information System. The 5-year pregnancy and birth outcome household survey helps to understand the magnitude of and district-level variations in maternal mortality for recent years, whereas the indirect sisterhood method of this paper is used to understand past maternal mortality level and variations in the region and in the districts included in the study.

Previous studies conducted in Ethiopia using the sisterhood method had limited areas of study. Moreover, the studies did not assess district-level variations in maternal mortality and failed to examine maternal mortality reduction over time. The indirect sisterhood method has been validated and used in different settings [9–12]. Our specific objectives in this study using the indirect sisterhood method were 1) to estimate the life time risk (LTR) of maternal death and the corresponding MMR; 2) to assess variations in maternal mortality estimates based on the characterstics of the respondents; and 3) to assess the reduction in MMR in Sidama Regional State over the past year by using the MMR of the sisterhood method (the main focus of this paper) with results from the 5-year recall of pregnancy and birth outcome household survey.

## Methods

### Study design and setting

We carried out a population-based cross-sectional study employing the sisterhood method (the main focus of this paper) as part of a larger maternal mortality household survey that used a retrospective 5-year recall of pregnancy and birth outcomes [8]. The study was conducted in Sidama National Regional State, Southern Ethiopia, in randomly selected 6 (20%) districts (Aleta Chuko, Aleta Wondo, Aroresa, Daela, Hawassa Zuriya, and Wondogenet) of the region. The study was conducted from July 2019 to May 2020. Sidama National Regional State is one of 11 regional states in the country, and its capital, Hawassa, is located 273 km south of Addis Ababa. The projected population of the region for 2020 was 4.3 million people. Administratively, the region is divided into 30 rural districts (*woredas*), 6 town administrations, and 536 rural *kebeles* (the smallest administrative structure with an average population of 5000 people). Under the *kebele*, there is a local structure known as *limatbudin* (a local administrative unit consisting of 40–50 neighbouring households on average).

Sidama Region has 18 hospitals (13 primary, 4 general, and 1 tertiary), 137 health centres, and 553 health posts run by the government [13]. In the region, there are also 4 hospitals (1 general and 3 primary), 21 specialty and higher clinics, 131 medium clinics, and 79 primary clinics operated by private owners (*Source*: *Sidama National Regional State Health Bureau unpublished report*, *2022*). Physical health service coverage of the region is 90.3% [14].

### The sisterhood method

The sisterhood method utilizes information obtained from adult men and women about the deaths of their sisters born to the same mother and who were of reproductive age. The sisterhood method uses the proportions of adult sisters dying during pregnancy, childbirth, or the puerperium reported by adults during a census or survey. In this way, a cohort of women of reproductive age at risk of pregnancy-related death is created [12, 15]. The LTR of maternal death is estimated from the sister unit of risk exposure to maternal mortality. MMR is approximated using the total fertility rate (TFR) of the study area [12, 15]. The MMR estimate from the indirect sisterhood method in average refers to 10–12 years prior to data collection and even can extend to 35 years when the respondents are older [12]. The sisterhood method is acclaimed for simplicity, minimal time, and inexpensiveness. However, it's not appropriate for monitoring short-term progress [16].

### Study population and sampling techniques

Men and women aged 15–49 years [11, 16, 17] in Sidama National Regional State were the source population of the study, and men and women of the same age and residing in sampled households were the study population.

This study is part of a larger study aimed at measuring maternal mortality, and the details of the sampling techniques of the study have been described elsewhere [8]. In the larger study, 8880 households with a corresponding number of mothers were needed to find 16,000 live births to estimate the maternal mortality ratio (MMR) in the region. Below, we summarize the sampling technique of the sisterhood method conducted as part of a larger maternal mortality household survey. Multistage cluster sampling technique was employed to identify the study participants. At the first stage, we randomly selected six districts from the region. At the second stage, we selected 40 *kebeles* from the six districts proportional to the size of the *kebeles* in the districts. Thirdly, from each *kebele*, we sampled 6 *limatbudin* and finally selected 37 households from each *limatbudin*. In the sampled households, the data collectors interviewed the

husband, wife, children, and any family member aged 15–49 years. When two or more eligible participants born to the same mother were found in the same household, one of them was chosen by a lottery method to avoid multiple counting.

## Variables

Life time risk of maternal deaths: the probability that a 15-year-old female will die eventually from a maternal cause and MMR; maternal deaths per 100,000 LB were the outcome measures of the study. We also collected characterstics of the respondents, like sex, age and educational level to assess their association with the outcome measures of interest.

## Data source and measurement

We carried out an interview with 17444 participants and analysed the data of 17374 participants; 8884 (51.1%) men and 8490 (48.9%) women 15–49 years old. The participants were residents of the 8880 households in the survey [8]. The data was collected by diploma- level teachers recruited from each *kebele* who were familiar with the culture and language of the study area. As this study was related to maternal health services provided by health professionals, we did not use health professionals as data collectors to minimize interviewers' bias. The data collectors conducted an interview with men and women aged 15–49 using interviewer-administered questionnaire. Participants were asked the following four standard questions used in the indirect sisterhood method [12]. 1) How many sisters (born to the same mother) have you ever had who reached 15 years old? 2) How many of these sisters who reached 15 years old are alive now? 3) How many of these sisters who reached 15 years old are dead? 4) How many of these dead sisters died while they were pregnant, during childbirth, or during the six weeks after the end of pregnancy?

The first question addresses the number of sisters who were at risk of pregnancy-related deaths, while the fourth question identifies the actual number of pregnancy-related deaths. Interviewers were observant to ensure the sum of questions 2 and 3 equals question 1 as a quality assurance of numbers. The period of six weeks after the end of pregnancy in question 4 was approximated to be two months after the end of pregnancy.

The age of the participants was obtained by directly asking the participants about their age and verified by the interviewers using key local and national events. A completed grade or education level was used to label the highest educational level attained, and those who could not read or write were labelled as having "no formal education".

## Data quality control

We adapted the standard sisterhood questions to the local context [12]. The questionnaire was prepared in English, translated into the local language (*Sidaamu Afoo*), and then back translated to English by another individual to check its consistency. Before actual data collection, the questionnaire was pretested in one district not included in the study.

The data collectors and supervisors were trained by the principal investigator. The training was part of the larger survey training, in which one day was scheduled for the sisterhood. The training content included discussion on the content and aim of each question, interview techniques, and role plays. The data collectors were supervised by two public health officers recruited from each district who were also familiar with the culture and language of the study setting. The supervisors followed the data collectors and checked the consistency and completeness of the questionnaire. When eligible respondents were absent during the initial visit, the data collectors revisited the households the next day.

The data was double-entered using EpiData software (EpiData Association 2000–2021, Denmark). The consistency of the two entries was checked, and discrepancies were validated from the hard copies of the questionnaire.

## Sample size determination

Hanley and colleagues suggested that, in settings where the MMR is within the range of 500 per 100,000 LB, the deaths of 385 sisters (± 10% margin of error) will be reported from interviews with 13000 adult respondents aged 15–49 [15]. According to the 2016 EDHS report, the MMR of Ethiopia was 412/100,000 LB [7]. Adding 10% of non-responses, we decided to interview 14,300 respondents. The number of households planned for the larger maternal mortality household survey was 8880 [8]. According to the World Health Organization guidelines on the sisterhood method, at least two adult respondents would be found in a household [16]. We assumed that each household would on average provide at least two adult respondents, so surveying 8880 households would be sufficient to get the desired sample of adult respondents for the sisterhood study.

## Statistical analysis

Stata version 15 was used for data analysis. Before LTR and MMR computation, two adjustments were made by grouping the participants' ages in five age groups to estimate the number of sisters who would be reported by the younger age group and to get the number of sisters exposed to full-time risk exposure at each age group [12]. The respondents in the younger age group (15–19 and 20–24) would have sisters who have not yet reached age 15. Hence, we created a hypothetical number of sisters that would have been reported by the young age group. To get this number, we calculated the average number of sisters reported by the older age group (25–49) and multiplied it by the initial number of sisters reported by the younger age group. The average number of sisters reported by age group 25–49 was 2.77 (37072/13401). Therefore, the number of sisters reported by age group 15–19 was 1475, and multiplied by 2.77, it equals 4086; and the same is true for age group 20–24 (8386 sisters * 2.77 = 23229).

The second adjustment was made to get sisters exposed to full lifetime risk exposure [12]. As we interviewed participants aged below 50, we did not have information about the full lifetime risk of all their sisters. Hence, we multiplied the sister unit of risk exposure of each age group with the adjustment factor in order to get "complete" sister units of risk exposure. The adjustment factor was a number calculated for the risk exposure of different age groups in developing countries, which is independent of a particular country [12].

Descriptive statistics with means and percentages were computed to describe participants' characteristics. The LTR for maternal death was obtained by dividing the total number of maternal deaths reported by the estimated total number of sisters exposed (LTR = total number of maternal deaths / total sister units of risk exposure). We used the total fertility rate (TFR) of the rural population of Ethiopia (5.2) [7] to estimate MMR from the LTR. (MMR = (1-[(1- LTR) 1/TFR] x 100,000), using formulas specified by Hanley et al. [15].

The corresponding time period to which our estimate refers was computed using the following formula: $T = \Sigma(T(i)*B(i))/\Sigma B(i)$, where T = the point time location of the global estimate, $T(i)$ = the time location of the estimate for each age group, and $B(i)$ = the exposing units of each age group [12]. We carried out stratified analysis based on respondents' sex, age, and location to see the association with the outcome measure.

Finally, we assessed differences and reductions in MMR in Sidama Regional State and in the districts included in this study using the two maternal mortality estimation methods: the sisterhood method and the 5-year recall of pregnancy and birth outcome household survey

[8]. The reference period for the sisterhood method MMR estimation was 2010, whereas the reference period for the 5-year recall of pregnancy and childbirth outcome MMR estimation was from July 2014 to June 2019.

## Ethical approval

We obtained ethical approval for this study from the institutional review board (IRB/015/11) of Hawassa University College of Medicine and Health Sciences and the Regional Ethical Committee of Western Norway (2018/2389/REK Vest). We got support letters from the Sidama Regional Health Bureau (formerly known as the Sidama Zone Health Department), and from the district health offices and kebele administrations. Informed (thumb print and singed) consent was obtained from the respondents. To maintain the anonymity of the study participants, we did not collect any personal identifying information during data collection.

## Inclusivity in global research

Additional information regarding the ethical, cultural, and scientific considerations specific to inclusivity in global research is included in the Supporting Information (S2 Checklist).

## Results

Table 1 shows the background characteristics of the respondents. We conducted an interview with 17,444 men and women 15–49 years of age residing in 8880 households. Seventy (0.4%) respondents had incomplete information and were excluded from the analysis. There were no

**Table 1. Characterstics of respondents of maternal mortality survey using the sisterhood method, in Sidama National Regional State, southern Ethiopia, 2020.**

| Characterstics of respondents | Number of respondents | Percentage |
|---|---|---|
| District (Woreda) | | |
| Aleta Chuko | 3722 | 21.4 |
| Aleta Wondo | 3929 | 22.6 |
| Aroresa | 2530 | 14.6 |
| Daela | 990 | 5.7 |
| Hawassa Zuriya | 3925 | 22.6 |
| Wondogenet | 2278 | 13.1 |
| Sex | | |
| Men | 8490 | 48.9 |
| Women | 8884 | 51.1 |
| Educational level | | |
| No formal education | 3557 | 20.5 |
| Grade 1–4 | 4397 | 25.3 |
| Grade 5–8 | 6776 | 39.0 |
| High school or above | 2644 | 15.2 |
| Age group | | |
| 15–19 | 706 | 4.1 |
| 20–24 | 3267 | 18.8 |
| 25–29 | 4892 | 28.1 |
| 30–34 | 4216 | 24.3 |
| 35–39 | 2530 | 14.6 |
| 40–44 | 1221 | 7.0 |
| 45–49 | 542 | 3.1 |

**Table 2. Estimate of lifetime risk of maternal death using the sisterhood method referring to 2010, in Sidama National Regional State, southern Ethiopia, 2020.**

| Age group of respondents (years) | Number of respondents (N) | Sisters reached age > = 15 (e) | Number of death from all cause (c) | Number of Pregnancy related death (r) | Adjustment factor (f) | Sister units of exposure (B) = ef | Lifetime risk (Q) = r/B |
|---|---|---|---|---|---|---|---|
| 15–19 | 706 | 4086* | 33 | 9 | 0.107 | 437 | 0.021 |
| 20–24 | 3267 | 23229* | 269 | 77 | 0.206 | 4785 | 0.016 |
| 25–29 | 4892 | 13474 | 561 | 191 | 0.343 | 4622 | 0.041 |
| 30–34 | 4216 | 11867 | 706 | 227 | 0.503 | 5969 | 0.038 |
| 35–39 | 2530 | 7123 | 464 | 159 | 0.664 | 4730 | 0.034 |
| 40–44 | 1221 | 3289 | 258 | 73 | 0.802 | 2638 | 0.028 |
| 45–49 | 542 | 1319 | 111 | 40 | 0.9 | 1187 | 0.034 |
| Total | 17374 | 64387 | 2402 | 776 | | 24368 | 0.032 |

Note:

* The expected number of sisters reaching age 15 and above in the younger age groups (age 15–19 and age 20–24) was calculated by multiplying the average number of sisters reported among respondents aged 25–49 (which is 2.77) with the original number of sisters reported by respondents in younger age. The original number of reported sisters in age group 15–19 was 1475 while in age group 20–24 was 8386

reported deaths of sisters among the 70 respondents with incomplete information. The final analysis included 17,374 (99.6%) respondents: 8,884 (51.1%) women and 8,490 (48.9%) men. The mean age of the respondents was 29.3 years (SD = 6.8), 20.5% had no formal education, and 15.2% had attended high school or higher education.

Table 2 shows maternal mortality estimates by 5 year age groups. The 17,374 respondents reported a total of 64,387 maternal sisters who had reached 15 years of age or older, and on average, each respondent reported 3.7 sisters. Of the 64,387 reported sisters who had reached 15 years of age or older, 2,402 (3.7%) had died. Of those who had died, 776 (32.3%) were pregnancy-related deaths. The total LTR of maternal death was 3.2%, i.e. 1 in 31 women 15–49 years old dies due to maternal causes. Using a total fertility rate of 5.2, the estimated MMR for the study area was 623 (95% CI: 573–658) per 100,000 LB. The approximate time reference for the MMR estimate was the year 2010, corresponding to 10 years prior to data collection.

To estimate MMR for the most recent period, we repeated the procedure in Table 2, stratifying the participants into three age groups: 15–29, 30–39, and 40–49 (Table 3). The MMR among participants in the 15–29 age group was 545 (95% CI: 475–601) per 100, 000 LB,

**Table 3. Maternal mortality estimate and reference period by age group using the sisterhood method, in Sidama National Regional State, southern Ethiopia, 2020.**

| Age group | 15–29 | 30–39 | 40–49 | Total |
|---|---|---|---|---|
| Respondents | 8865 | 6746 | 1763 | 17374 |
| Sisters 15y+ | 40789 | 18990 | 4608 | 64387 |
| Sisters 15y+ died | 863 | 1170 | 369 | 2402 |
| Sisters 15y+ died maternal | 277 | 386 | 113 | 776 |
| Sisters exposed to risk | 9844 | 10699 | 3825 | 24368 |
| Lifetime risk | 0.028 | 0.036 | 0.029 | 0.032 |
| MMR* | 545 | 703 | 564 | 623 |
| 95% CI for MMR | 476–601 | 624–760 | 455–660 | 573–658 |
| Estimated years back | 7.4 | 10.6 | 15.3 | 10.2 |

Note:

* Maternal mortality ratio per 100, 000 LB, CI: Confidence Interval

**Table 4. Maternal mortality estimate by sex of respondents, using the sisterhood method, in Sidama National Regional State, southern Ethiopia, 2020.**

|  | Male | Female | Total |
|---|---|---|---|
| Respondents | 8490 | 8884 | 17374 |
| Sisters 15y+ | 26087 | 38485 | 64387 |
| Sisters 15y+ died | 1276 | 1126 | 2402 |
| Sisters 15y+ died maternal | 397 | 379 | 776 |
| Sisters exposed to risk | 12830 | 11574 | 24368 |
| Lifetime risk | 0.031 | 0.033 | 0.032 |
| MMR* | 604 | 643 | 623 |
| 95% CI for MMR | 538–654 | 572–697 | 573–658 |

Note:

* Maternal mortality ratio per 100,000 live birth, CI: Confidence Interval

reflecting 7 years before data collection. For respondents in the 30–39 age group, the MMR was 703 (624–760) per 100, 000 LB, reflecting a period of 11 years before the study. For respondents 40 years of age or older, the MMR was 564 (455–660) per 100, 000 LB, reflecting a period of 15 years before data collection.

Table 4 shows the life-time risk of maternal death and MMR stratified by male and female respondents. There was no statistically significant difference among male and female respondents.

Table 5 shows a stratified analysis of maternal mortality estimates by the districts of respondents. The MMR in Aroresa district was significantly higher than all other study districts in the region: MMR: 1210 (95% CI: 1027–1318) per 100,000 LB. Aroresa district is the most remote district, located 181 km away from the regional centre, Hawassa, followed by Daela district, which is situated 175 km away from Hawassa.

Table 6 describes the estimated MMR by two different methods. The sisterhood method refers to around 10 years before the study and showed an overall MMR of 623/100,000 LB in Sidama Regional State. The survey with a 5-year recall found 419/100,000 LB with slightly overlapping confidence intervals. Some districts had a lower MMR in the survey of 5-year pregnancy recall than the sisterhood estimates reflecting 10 years ago, whereas others, like Aroresa, had similar estimates in both.

**Table 5. Maternal mortality estimate by district of respondents using the sisterhood method, in Sidama National Regional State, southern Ethiopia, 2020.**

|  | Aleta- Chuko | Aleta-Wondo | Aroresa | Daela | Hawassa Zuriya | Wondo-genet |
|---|---|---|---|---|---|---|
| Respondents | 3722 | 3929 | 2530 | 990 | 3925 | 2278 |
| Sisters 15y+ | 13602 | 17155 | 9924 | 4107 | 12557 | 6962 |
| Sisters 15y+ died | 595 | 478 | 502 | 171 | 441 | 215 |
| Sisters 15y+ died maternal | 136 | 128 | 236 | 48 | 161 | 67 |
| Sisters exposed to risk | 5432 | 5949 | 3848 | 1725 | 4446 | 2938 |
| Lifetime risk | 0.025 | 0.022 | 0.061 | 0.028 | 0.036 | 0.023 |
| MMR* estimate | 486 | 417 | 1210 | 541 | 707 | 442 |
| 95% CI for MMR | 409–561 | 351–495 | 1027–1318 | 389–688 | 587–798 | 338–547 |
| Distance from region, Hawassa | 61 km | 64 km | 181 km | 175 km | 21 km | 25 km |

Note:

* Maternal mortality ratio per 100, 000 LB, CI: Confidence Interval, km: kilo meter

**Table 6. Estimates of maternal mortality ratio by two methods in Sidama National Regional State, southern Ethiopia, 2020.**

| | MMR using 5 year recall of pregnancy and birth outcome* | | | MMR using the indirect sisterhood method** | | | | |
|---|---|---|---|---|---|---|---|---|
| **District** | **Live birth** | **Maternal death** | **MMR (95% CI)** | **Number of respondents** | **Pregnancy related deaths** | **Sister units exposed to risk** | **LTR** | **MMR (95% CI)** |
| Aleta Chuko | 2265 | 5 | 263 (58–467) | 3722 | 136 | 5432 | 0.025 | 486 (409–561) |
| Aleta Wondo | 2285 | 12 | 525 (207–842) | 3929 | 128 | 5949 | 0.022 | 417 (351–495) |
| Aroresa | 1694 | 21 | 1142 (693–1591) | 2530 | 236 | 3848 | 0.061 | 1210 (1027–1318) |
| Daela | 716 | 4 | 641 (77–1358) | 990 | 48 | 1725 | 0.028 | 541 (389–688) |
| Hawassa-Zuriya | 2237 | 2 | 114 (24–251) | 3925 | 161 | 4446 | 0.036 | 707 (587–798) |
| Wondogenet | 1405 | 4 | 258 (8–508) | 2278 | 67 | 2938 | 0.023 | 442 (338–547) |
| Sidama Region | 10602 | 48 | 419 (260–577) | 17374 | 776 | 24338 | 0.032 | 623 (573–658) |

Note:

*Reference year: July 2014-June 2019,

** Reference year: 2010, Abbreviations: MMR: maternal mortality ratio per 100,000 live birth, LTR: life time risk for maternal death, CI: confidence interval

## Discussion

### Principal findings

By incorporating the sisterhood method in a household survey of pregnancy and birth outcomes, we found a lifetime risk of maternal death of 3.2%, with a corresponding MMR of 623 per 100,000 LB; the time reference was 2010. The remote district (Aroresa) had a significantly higher MMR. A sub analysis of MMR based on the respondents' age showed that participants 15–29 years of age had a MMR; 545 per 100, 000 LB, reflecting 7 years before data collection. A stratified analysis for male and female respondents provided a similar maternal mortality estimate.

The MMR estimated by a household survey that used a 5-year recall of pregnancy found an estimated MMR of 419 per 100,000 LB, and the sisterhood method, referring to 10 years before the study, found 623 per 100,000 LB, with slightly overlapping confidence intervals. Districts located far from the centre with poor infrastructure and inadequate skilled health personnel did have similar MMR by both methods and seem to have persistently high MMR. Whereas, districts located near the centre with good infrastructure and adequate skilled health personnel have a lower estimated MMR by the 5-year recall than by the sisterhood method.

### Strengths and weaknesses of the study

To the best of our knowledge, this is the first study describing maternal mortality estimates employing the sisterhood method in Sidama National Regional State, southern Ethiopia. The study was conducted using a large and representative sample of the region that demonstrated district-level variations. By assessing the MMR of the sisterhood method and the estimates from a household maternal mortality survey that used 5-year pregnancy and birth outcomes [8], this study identified variations in MMR reduction in different geographical areas of the region.

This study had some limitations similar to those of other studies that employed the sisterhood method. The maternal mortality estimates we reported refer to a period around 10 years before data collection; thus, we were not able to show recent estimates using this method.

There might be over-reporting of deaths. Over-reporting of deaths may arise due to the inclusion of deaths that occurred beyond six weeks of the end of pregnancy or cases that were

not related to pregnancy. Multiple counting could also be the reason for the over-estimation of maternal deaths using the sisterhood method.

To minimize over reporting, we used data collectors who were familiar with the language and culture of the study population. As a result of familiarity with the local context, the data collectors supported participants in identifying the pregnancy state of deceased sisters and the time of their deaths. In addition, in households where more than one eligible participant born to the same mother was found, we chose one of them by the lottery method, which minimized multiple counting. Despite our effort, there might be over reporting in our study.

Underreporting could also be another limitation for this study. Deaths occurring at an early stage of pregnancy due to abortion or ectopic pregnancy and deaths of women not in marital union might not be reported.

We did not have information on the place of residence of the sisters in the cohort. We used the respondents' residences as a proxy for the sisters' location. Some sisters might have moved to other districts. Our study also lacks information on the age at death of respondents' sisters, but we used respondents' ages to observe recent deaths and show patterns of deaths across age groups. The age of the respondents may not closely reflect the age of their sisters.

## Maternal mortality ratios and variations using the sisterhood method

We found a high MMR in the study area with a time reference of 2010. Our result is significantly higher than the findings of the national DHS [7], which reported a MMR of 412 per 100,000 LB (time reference, 2009–2016). The DHS study uses the direct sisterhood method, which included detailed information on age at death, year of deaths, and years since the death occurred, which contributed to precise maternal mortality estimates [18]. However, in our study, we used the indirect sisterhood method, in which such detailed information is not included. The inherent weakness of the indirect sisterhood method is that it lacks exact information on age at death, year of death, and years since the death occurred, which may overestimate the deaths [16].

The estimate from our study was also higher than a study from Kersa Health and Demographic Surveillance Site (HDSS) [5], which estimated a MMR of 396 per 100,000 LB for the year 2010. The variation could be explained in the differences of methods the two studies used for MMR estimation, access to maternal health care, and differences in documentation of maternal deaths. Kersa's study employed the direct method of maternal mortality estimation, whereas we used the indirect sisterhood method, which might have caused over reporting of maternal deaths [11]. Mothers in Kersa DHSS might have better access to health-related information and improved maternal health service utilization due to DHSS activities, which in turn might have contributed to the reduction of maternal deaths in the area. In addition, in Kersa's DHSS, there might also be an improved vital registration system that supports accurate documentation of maternal deaths.

A study from Gamo-Gofa, south-west Ethiopia, using the sisterhood method with a time reference of 1998 found a MMR of 1667 per 100,000 LB, which is higher than our study [11] but carried out a decade before our study. The government has implemented many interventions since that study to improve universal health coverage for maternal health services, which might have contributed to a reduction in maternal mortality in our area [1].

In our study, we found similar maternal mortality estimates from both male and female respondents. This shows that the information obtained from male siblings is as valid as that obtained from female siblings in estimating maternal mortality using the indirect sisterhood method. This is in agreement with a study report from southwest Ethiopia [11]. A study carried out using DHS data from African countries also found similar findings [17].

This study found high maternal mortality estimates among the respondents in the 30–39 age group (MMR: 703 per 100,000 LB) compared to participants in the 15–29 age group. Our finding is different from findings reported from Nigeria, where they found the highest MMR among the respondents in the 15–29 age group [10]. Though we did not have information on the age of the diseased sisters (only respondents), the difference in age at first marriage might have contributed to the variations observed in two settings. In Ethiopia, the average age at first marriage was reported to be 17 [7], while 13 years was reported in Mali [9].

Our study has also shown significant variations in MMR by districts among the respondents. This is in agreement with the findings of a study conducted in northern Nigeria and Mali [9, 10], where they reported that maternal mortality differed by local government areas and villages, respectively. The areas with high maternal mortality in those studies were characterized by remoteness and problematic access to health facilities. Previously, studies have used the place of the respondents as a proxy location for the sisters of the respondents to describe the differences in deaths based on the location of the respondents [9, 10].

The high MMR observed in Aroresa district (the remote district) could be explained by poor road facilities and difficult topography that might hamper access to health services. Aroresa district is situated 181 km away from the regional centre [19]. The distance from the centre may affect the availability of services in an emergency situation. Weak emergency obstetric care, compounded by a lack of adequate and skilled personnel, might have also contributed to the high maternal mortality in the district. The low utilization rate of maternal health services could also explain the high MMR in Aroresa district, and a paper from this area in 2018 reported institutional delivery at 38% and use of contraceptives at 49% [19].

## Maternal mortality reduction in Sidama National Regional State

A study from India reported that there were regional and district level variations in maternal mortality reduction across the country [20]. In our study of Hawassa Zuriya district (the central district), the sisterhood study reflecting 10 years ago gave a MMR of 707, and the 5-year recall gave 114 per 100.000 LB. The apparent reduction in MMR in this district could be attributed to improved access to obstetric care during emergency situations as the district is located near the referral centre at the regional capital, Hawassa [8], with good infrastructure and road facilities. Moreover, the district is characterized by having a relatively high doctor-to-population ratio and a midwife-to-population ratio which might have improved access to skilled obstetric care [8]. An Indonesian study demonstrated that investment in doctors and hospitals would significantly reduce the MMR [21]. Contrary to Hawassa Zuriya district (the central district), in Aroresa district (the remote district), the two studies found high MMR with no reduction of MMR over the past years.

## Validity of maternal mortality measurements

We incorporated the sisterhood questions into the ongoing maternal mortality household survey that used a 5-year recall of pregnancy and birth outcomes to determine the MMR. We added four questions of the indirect sisterhood to the main study questionnaire and collected the data from respondents residing in the same households sampled for the main study using the same enumerators. There was no major cost or logistics incurred due to the inclusion of the sisterhood questions in the main household survey questionnaire. However, the main household study incurred costs in terms of time, logistics, and money. Despite the costs incurred, such studies provide important information on the magnitude and variations of MMR, which is useful for program planning, priority setting, and resource allocation at a lower administrative level.

We considered the size of the sample and sampling techniques while conducting the studies using the two maternal mortality estimation methods in order to precisely estimate the MMR in the region. Hence, a representative and large number of participants were included in both the 5-year recall of pregnancy and birth outcome household survey and the sisterhood study. In the 5-year recall of pregnancy and birth outcome household survey, we registered 10602 LB and 48 maternal deaths in 8880 households visited. Our aim was to find 66 maternal deaths with MMR, 412 (95% CI: 324–524) per 100,000 LB. The MMR after the study was 419 (95% CI: 260–577), which is within the 95% CI we anticipated initially [8]. For the sisterhood study, we had an interview with 17444 siblings. This sample size was above the recommended 13,000 siblings for the indirect sisterhood study in settings with similar magnitudes of maternal mortality [15]. The samples for both studies were selected using probability sampling, and a multi-stage cluster sampling technique was employed to select the study participants.

The MMR we found using the sisterhood method was higher than the findings from other studies in the same time reference [7]. Overestimation of MMR from studies that used the sisterhood method have been documented [11]. The MMR obtained from the household survey was comparable with other national figures [2, 7].

Recall bias is a potential limitation both in the household survey study and the sisterhood study, as the conclusions of the studies were based on the information obtained from families memorizing past events of maternal deaths. We believe that the recall bias is lower in the 5-year recall of pregnancy and birth outcome study than in the sisterhood as the deaths occurred in recent years and the interviews were conducted with close relatives who lived together with the deceased mother.

Underestimation could also be another limitation for both studies. The stigma of pregnancies among unmarried girls may lead to underreporting. Maternal deaths occurring at an early stage of pregnancy due to abortion or ectopic pregnancy may be missed and regarded as non-maternal deaths within the community.

### Policy and clinical implications

Persistently high maternal mortality in districts found in remote areas far away from central city with poor infrastructure and inadequate skilled health workers calls for attention from the government and the importance of instituting interventions tailored to the local context to improve poor infrastructure and the shortage of skilled health personnel in Sidama Region State. Analysis of results from recent maternal mortality estimation using the 5-year recall of pregnancy and birth outcome household survey with the sisterhood method suggests the possibility of monitoring maternal mortality reduction at regional and district level using the two estimation methods.

### Conclusions

This study used two methods for estimating maternal deaths in Sidama, Ethiopia, one referring to the last 5 years and the other 10 years ago. The central districts have a lower MMR than before, but peripheral districts still have a high MMR. The Sidama Regional Health Bureau and respective district health offices should implement interventions tailored to the local context to address variations and accelerate reductions in MMR.

### Supporting information

**S1 Checklist. STROBE statement.**
(DOCX)

**S2 Checklist. Inclusivity in global research.**
(DOCX)

## Acknowledgments

We are grateful to the participants in the study for their time and information. Our special gratitude goes to the Sidama National Regional Health Bureau, respective district health offices, and *kebele* administrations in Ethiopia for their support and permission to conduct the study.

## Author Contributions

**Conceptualization:** Aschenaki Zerihun Kea, Bernt Lindtjorn, Sven Gudmund Hinderaker.

**Data curation:** Aschenaki Zerihun Kea.

**Formal analysis:** Aschenaki Zerihun Kea.

**Investigation:** Aschenaki Zerihun Kea, Bernt Lindtjorn, Sven Gudmund Hinderaker.

**Methodology:** Aschenaki Zerihun Kea, Bernt Lindtjorn, Sven Gudmund Hinderaker.

**Project administration:** Aschenaki Zerihun Kea.

**Supervision:** Bernt Lindtjorn, Achamyelesh Gebretsadik Tekle, Sven Gudmund Hinderaker.

**Writing – original draft:** Aschenaki Zerihun Kea.

**Writing – review & editing:** Aschenaki Zerihun Kea, Bernt Lindtjorn, Achamyelesh Gebretsadik Tekle, Sven Gudmund Hinderaker.

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
