## [Decision Letter · Decision Letter 0]

18 Jul 2023

PONE-D-22-25845Reduction in maternal mortality ratio varies by district in Sidama Regional State, southern Ethiopia: Estimates by cross-sectional studies using the sisterhood method and a household survey of pregnancy and birth outcomesPLOS ONE

Dear Dr.Aschenaki Zerihun

Thank you for submitting your manuscript to PLOS ONE. After careful consideration, we feel that it has merit but does not fully meet PLOS ONE’s publication criteria as it currently stands. Therefore, we invite you to submit a revised version of the manuscript that addresses the points raised during the review process.

We look forward to receiving your revised manuscript.

Kind regards,

Anteneh Fikrie, MPH

Academic Editor

PLOS ONE

Journal Requirements:

3. Please include a complete copy of PLOS’ questionnaire on inclusivity in global research in your revised manuscript. Our policy for research in this area aims to improve transparency in the reporting of research performed outside of researchers’ own country or community. The policy applies to researchers who have travelled to a different country to conduct research, research with Indigenous populations or their lands, and research on cultural artefacts. The questionnaire can also be requested at the journal’s discretion for any other submissions, even if these conditions are not met.  Please find more information on the policy and a link to download a blank copy of the questionnaire here: https://journals.plos.org/plosone/s/best-practices-in-research-reporting. Please upload a completed version of your questionnaire as Supporting Information when you resubmit your manuscript.

Reviewers' comments:

Reviewer's Responses to Questions

**Comments to the Author**

1. Is the manuscript technically sound, and do the data support the conclusions?

Reviewer #1: Yes

Reviewer #2: Yes

2. Has the statistical analysis been performed appropriately and rigorously? 

Reviewer #1: Yes

Reviewer #2: Yes

3. Have the authors made all data underlying the findings in their manuscript fully available?

Reviewer #1: Yes

Reviewer #2: Yes

4. Is the manuscript presented in an intelligible fashion and written in standard English?

Reviewer #1: Yes

Reviewer #2: Yes

5. Review Comments to the Author

Reviewer #1: Reduction in maternal mortality ratio varies by district in Sidama Regional State, southern Ethiopia: Estimates by cross-sectional studies using the sisterhood method and a household survey of pregnancy and birth outcomes

Abstract section

1.Under background section you said that “Maternal mortality studies conducted at national level do not provide estimates useful for monitoring maternal mortality at lower administrative level.” Do you think this is the reason to conduct this study? I don’t think? The gap of study is not in line with the aim, please check it?

2.Under method section: The study was conducted with men and women aged 15- 27 49 years, why you include men? Under this section it is better to add how you analysis the data?

3.How many woreads were included in your study? Do your intention to make comparison between remote district and town admiration? What do you mean remote district? How many district were remote? What is the implication of this comparison? Why you did not compare urban and rural district?

4.Under conclusion section: “The high MMR with district level variations” How many district were compare to show variations?

Introduction

1.Line 49-50: “Despite these achievements, in 2017, there were an estimated 14000 maternal deaths with corresponding maternal mortality ratio (MMR) 401 per 100,000 live births (LB). From where you get this, I did not see any references? Also you reported 2017 which is old there are a lot of change in Ethiopia after this period, why you show the updated figures?

2.From line 63 to 73: you describe about indirect sisterhood method to measure maternal mortality. What make this study from previously conducted study? Please clear show as the gap what this study will add?

Method section

3.Line 114-115: you said that “Thirdly, from each kebele, we sampled 6 limatbudin and finally selected 37 households from each limatbudin” why 6 limatbudin? Why 37 households? Add references?

4.Sample size: your calculated sample size after adding 10% non-responses was 14,300 respondents. But, you assume 8880 household are sufficient? What was the base for your assumptions? For me this is difficult to accept your assumptions?

5.In line 174 : you said that 2024, what is 2024 indicated? Do you mean the age between 20 to 24? Check it

6.Why you did not include those aged between 10 to 14 years?

7.How we know Aroresa district is the remote district, what about Aleta Chuko? Daela?

8.Generalizability issues? Can we make generalizability for urban and rural districts?

Reviewer #2: Review Report

I would like to thank the authors for submitting their manuscript to PLOS ONE and for the opportunity to review it.

The manuscript presents the results of population based cross-sectional study assessing the variations in reduction of maternal mortality ratio in Sidama Regional State, southern Ethiopia. The manuscript is well written and comprehensible. Adequate statistical methods were used. These are my further comments/suggestions:

•Key words were missing from the abstract of the paper, although included in the system.

•The introduction section is well written but only 12 studies were reviewed, the authors should think of reviewing various global and local studies.

•The description of the study setting lacks some important factors of the region which can be related to MM like the topography, literacy status, health service coverage....

•Minor editorial issue in page 6 line number 103, remove the s from the word appropriates

•What was done if any of the selected household lacks an eligible participant? It should be written.

•The reason why the authors selected Diploma level teachers as data collectors is not clear for me; would it have been better if they selected health professionals?

•Number of men and women study participants reported in the result description and reported in the table 1 were different? Needs revision.

6. PLOS authors have the option to publish the peer review history of their article (what does this mean?). If published, this will include your full peer review and any attached files.

Reviewer #1: **Yes: **Negussie Boti Sidamo

Reviewer #2: No

---

## [Author Response · Author response to Decision Letter 0]

26 Aug 2023

PONE-D-22-25845

Reduction in maternal mortality ratio varies by district in Sidama Regional State, southern Ethiopia: Estimates by cross-sectional studies using the sisterhood method and a household survey of pregnancy and birth outcomes.

PLOS ONE

Point-by-point response to reviewers’ and editor comments

The authors would like to thank the reviewers and editor for their time, suggestions, and valuable comments. We have carefully addressed all the comments and suggestions. The corresponding changes made to the revised manuscript are summarized in our response below.

Response to Reviewer #1 comments 

1. Abstract Section 

1.1. Background 

Comment: Under background section you said that “Maternal mortality studies conducted at national level do not provide estimates useful for monitoring maternal mortality at lower administrative level.” Do you think this is the reason to conduct this study? I don’t think? The gap of study is not in line with the aim, please check it? 

Answer: We would like to thank the reviewer for pointing out this issue. We have revised the sentence: “Few studies assess the magnitude, variations, and reduction of maternal mortality at a lower administrative level” (lines 20-21)

1.2. Method 

Comment: The study was conducted with men and women aged 15- 27 49 years, why you include men? Under this section it is better to add how you analysis the data? 

Answer: We acknowledge the question raised by the reviewer. Previous studies employed both men and women in the sisterhood study [Refs: 11, 16, and 17] (line 114). Some studies found comparable estimates of maternal mortality, some different findings. Hence, we wanted to assess maternal mortality estimates using both sexes. As the sisterhood method has a unique analysis, we described our analysis in the methods section of the abstract (lines 26-28). Other additional analysis conducted are also mentioned (lines 28-31).

Comment: How many woreads were included in your study? Do your intention to make comparison between remote district and town admiration? What do you mean remote district? How many district were remote? What is the implication of this comparison? Why you did not compare urban and rural district?

Answer: We appreciate the comment forwarded by the reviewer. In our study, there were 30 rural woredas (districts). In this study, six districts were randomly selected (line 24). All the districts included in our study were rural, so we could not compare urban and rural. However, among the 6 districts selected, 2 were remote, based on the distance from the regional center, Hawassa. We mentioned this in the result section (lines 274-276). We also included the distance the districts have from the regional center, Hawassa, in Table 3C. As mentioned in Table 4, Aroresa and Daela districts, the most remote, had the highest maternal mortality according to the results of the pregnancy and birth outcomes household survey. The two districts also showed a low reduction in maternal mortality in the past two decades. This shows the Sidama National Regional State Health Bureau, district health offices, and other concerned bodies should give due attention to the districts located far away from the regional center to reduce the high maternal mortality. 

Conclusion 

Comment: The high MMR with district level variations” How many district were compare to show variations?

Answer: We thank the reviewer for the question. We studied a representative sample of six districts in the region and made comparisons among them to see variations. 

2. Introduction Section 

Comment: Line 49-50: “Despite these achievements, in 2017, there were an estimated 14000 maternal deaths with corresponding maternal mortality ratio (MMR) 401 per 100,000 live births (LB). From where you get this, I did not see any references? Also you reported 2017 which is old there are a lot of change in Ethiopia after this period, why you show the updated figures?

Answer: We appreciate the reviewer pointing out missing references, they are now included [Ref. 2, line 50]. We acknowledge that a lot of things have changed in Ethiopia since 2017, and when we planned this study, this was the latest data we had. The statement in the introduction points out the situation when we planned the study, not after the study. 

Comment: From line 63 to 73: you describe about indirect sisterhood method to measure maternal mortality. What make this study from previously conducted study? Please clear show as the gap what this study will add?

Answer: We appreciate the reviewer for this question. Previous studies did not compare district-level variations in maternal mortality and did not assess maternal mortality reduction over time. We included a paragraph indicating the gap this study will fill (lines 73–75). We also used the findings of the larger study, of which this study was part, to assess the reduction in maternal mortality in the area. 

Method Section 

Comment: Line 114-115: you said that “Thirdly, from each kebele, we sampled 6 limatbudin and finally selected 37 households from each limatbudin” why 6 limatbudin? Why 37 households? Add references?

Answer: This study was part of a larger study aiming at measuring maternal mortality in the Sidama Region. For the sample size of the larger study, the total number of households needed was 8880.We decided to study 40 kebeles proportional to the size of the kebeles in the districts. We also decided to study six limatbudins from each kebele, which gives a total of 240 limatbudins (40*6). Finally, we sampled 37 households from each limatbudin, which equals a total of 8880 households (240*37). We have cited a reference for this [Ref. 8, line 118]. We have also added the description we mentioned above (line 117-121), and this has also been elaborated in the sample size calculation section of the manuscript. 

Comment: Sample size: your calculated sample size after adding 10% non-responses was 14,300 respondents. But, you assume 8880 household are sufficient? What was the base for your assumptions? For me this is difficult to accept your assumptions?

Answer: This study was part of a larger study, and the assumptions and details of the sample size calculation have been described in [Ref, 8]. In the World Health Organization guidelines for the sisterhood method, two adult respondents would be found in a household. We missed citing the reference now included in the manuscript [Ref: 16, lines 181-184]. Though we calculated the sample size for this study (14300 participants), based on the above assumption, we considered that the sample of households needed for the larger study (8880 households) would be adequate for the current study.

Comment: In line 174: you said that 2024, what is 2024 indicated? Do you mean the age between 20 to 24? Check it

Answer: Good point. We corrected as 20-24, line 191.

Comment: Why you did not include those aged between 10 to 14 years?

Answer: The standard sisterhood method guidelines and other studies conducted following the guidelines recommend the use of the 15–49 age group. We cited the references [Ref: 11, 16, and 17, line 114]. 

Comment: How we know Aroresa district is the remote district, what about Aleta Chuko? Daela?

Answer: We used the distance the districts have from the regional centre. We indicated this in Table 3C and described it in a sentence above the table (line 274-276).

Comment: Generalizability issues? Can we make generalizability for urban and rural districts?

Answer: As this study employed a random sample of districts in the region (both from rural and semi -urban areas), we can generalize the findings to the whole region as well as to similar settings in other areas.

Response to Reviewer #2 comments 

Comment: Key words were missing from the abstract of the paper, although included in the system

Answer: We appreciate the question raised by the reviewer. We have checked the PLOS ONE submission guidelines. We confirmed that key words are not included in the abstract section. Hence, we submitted separately in the submission system.

Comment: The introduction section is well written but only 12 studies were reviewed, the authors should think of reviewing various global and local studies.

Answer: Good suggestions. Our references include both global and national studies, in spite of their limited numbers.

Comment: The description of the study setting lacks some important factors of the region which can be related to MM like the topography, literacy status, health service coverage....

Answer: We would like to thank the reviewer for the suggestion. We added some lines about the number and type of health facilities and health service coverage of the study area (lines 96-100). We couldn’t find published data on literacy and the topography of the region. 

Comment: Minor editorial issue in page 6 line number 103, remove the s from the word appropriates

Answer: Edited, line 111 

Comment: What was done if any of the selected household lacks an eligible participant? It should be written.

Answer: If an eligible participant was not present in a household during the initial visit, the data collectors re-visited the household the next day (lines 170-171). We did not have households with no participants, as the sampled households were selected randomly from those households where eligible participants were residing. 

Comment: The reason why the authors selected Diploma level teachers as data collectors is not clear for me; would it have been better if they selected health professionals?

Answer: In this study, we did not use health professionals as data collectors to minimize the interviewer's bias because the study is related to maternal health services provided by the health professionals (lines 141-143). We preferred to use diploma-level teachers who were familiar with the culture and language of the study population. This time, every locality in the study area has a school, the teachers are easily available, and we assume their education level is sufficient to understand the questionnaire and properly collect the data if they are properly trained.

Comment: Number of men and women study participants reported in the result description and reported in the table 1 were different? Needs revision.

Answer: Thanks for pointing out the mistake. We have made correction, line 238.

---

## [Editor Report · Decision Letter 1]

28 Sep 2023

Reduction in maternal mortality ratio varies by district in Sidama National Regional State, southern Ethiopia: Estimates by cross-sectional studies using the sisterhood method and a household survey of pregnancy and birth outcomes

PONE-D-22-25845R1

Dear Dr. Zerihun,

We’re pleased to inform you that your manuscript has been judged scientifically suitable for publication and will be formally accepted for publication once it meets all outstanding technical requirements.

Kind regards,

Anteneh Fikrie, MPH

Academic Editor

PLOS ONE
---

## [Editor Report · Acceptance letter]

3 Oct 2023

PONE-D-22-25845R1 

Reduction in maternal mortality ratio varies by district in Sidama National Regional State, southern Ethiopia: Estimates by cross-sectional studies using the sisterhood method and a household survey of pregnancy and birth outcomes 

Dear Dr. Kea:

I'm pleased to inform you that your manuscript has been deemed suitable for publication in PLOS ONE. Congratulations! Your manuscript is now with our production department. 

Kind regards, 

on behalf of

Professor Anteneh Fikrie 

Academic Editor

PLOS ONE